# Individual variations in 'brain age' relate to early-life factors more than to longitudinal brain change

Didac Vidal-Pineiro[1]*, Yunpeng Wang[1], Stine K Krogsrud[1], Inge K Amlien[1], William FC Baaré[2], David Bartres-Faz[3], Lars Bertram[1,4], Andreas M Brandmaier[5,6], Christian A Drevon[7], Sandra Düzel[6], Klaus Ebmeier[8], Richard N Henson[9], Carme Junqué[3,10], Rogier Andrew Kievit[9,11], Simone Kühn[12,13], Esten Leonardsen[1], Ulman Lindenberger[5,6], Kathrine S Madsen[2,14], Fredrik Magnussen[1], Athanasia Monika Mowinckel[1], Lars Nyberg[15], James M Roe[1], Barbara Segura[3,10], Stephen M Smith[6], Øystein Sørensen[1], Sana Suri[16,17], Rene Westerhausen[18], Andrew Zalesky[19], Enikő Zsoldos[17], Kristine Beate Walhovd[1,20], Anders Fjell[1,20]

[1]Center for Lifespan Changes in Brain and Cognition, Department of Psychology, University of Oslo, Oslo, Norway; [2]Danish Research Centre for Magnetic Resonance, Centre for Functional and Diagnostic Imaging and Research, Copenhagen University Hospital - Amager and Hvidovre, Copenhagen, Denmark; [3]Department of Medicine, Faculty of Medicine and Health Sciences, Institute of Neurosciences, University of Barcelona; Institute of Biomedical Research August Pi i Sunyer (IDIBAPS), Barcelona, Spain; [4]Lübeck Interdisciplinary Platform for Genome Analytics (LIGA), University of Lübeck, Lubeck, Germany; [5]Max Planck UCL Centre for Computational Psychiatry and Ageing Research, Berlin, Germany; [6]Center for Lifespan Psychology, Max Planck Institute for Human Development, Berlin, Germany; [7]Department of Nutrition, Inst Basic Med Sciences, Faculty of Medicine, University of Oslo & Vitas Ltd, Oslo, Norway; [8]Department of Psychiatry, University of Oxford, Oxford, United Kingdom; [9]MRC Cognition and Brain Sciences Unit and Department of Psychiatry, University of Cambridge, Cambridge, United Kingdom; [10]Centro de Investigación Biomédica en Red sobre Enfermedades Neurodegenerativas (CIBERNED), Barcelona, Spain; [11]Cognitive Neuroscience Department, Donders Institute for Brain, Cognition and Behavior, Radboud University Medical Center, Nijmegen, Netherlands; [12]Lise Meitner Group for Environmental Neuroscience, Max Planck Institute for Human Development, Berlin, Germany; [13]Department of Psychiatry, University Medical Center Hamburg-Eppendorf, Hamburg, Germany; [14]Radiography, Department of Technology, University College Copenhagen, Copenhagen, Denmark; [15]Umeå Centre for Functional Brain Imaging, Department of Integrative Medical Biology, Physiology Section and Department of Radiation Sciences, Diagnostic Radiology, Umeå University, Umeå, Sweden; [16]Wellcome Centre for Integrative Neuroimaging (WIN FMRIB), University of Oxford, Oxford, United Kingdom; [17]Wellcome Centre for Integrative Neuroimaging, Departments of Psychiatry and Clinical Neuroscience, University of Oxford, Oxford, United Kingdom; [18]Section for Cognitive Neuroscience and Neuropsychology, Department of Psychology, University of Oslo, Oslo, Norway; [19]Department of Biomedical Engineering, Faculty of Engineering and IT, The University of Melbourne, Melbourne, Australia; [20]Department of radiology and nuclear medicine, Oslo University Hospital, Oslo, Norway

*For correspondence:
d.v.pineiro@psykologi.uio.no

**Abstract** *Brain age* is a widely used index for quantifying individuals' brain health as deviation from a normative brain aging trajectory. Higher-than-expected *brain age* is thought partially to reflect above-average rate of brain aging. Here, we explicitly tested this assumption in two independent large test datasets (UK Biobank [main] and Lifebrain [replication]; longitudinal observations ≈ 2750 and 4200) by assessing the relationship between cross-sectional and longitudinal estimates of *brain age*. *Brain age* models were estimated in two different training datasets (n ≈ 38,000 [main] and 1800 individuals [replication]) based on brain structural features. The results showed no association between cross-sectional *brain age* and the rate of brain change measured longitudinally. Rather, *brain age* in adulthood was associated with the congenital factors of birth weight and polygenic scores of *brain age,* assumed to reflect a constant, lifelong influence on brain structure from early life. The results call for nuanced interpretations of cross-sectional indices of the aging brain and question their validity as markers of ongoing within-person changes of the aging brain. Longitudinal imaging data should be preferred whenever the goal is to understand individual change trajectories of brain and cognition in aging.

## Introduction

The concept of *brain age* is increasingly used to capture interindividual differences in the structure, function, and neurochemistry of the aging brain (*Cole and Franke, 2017*). The biological age of the brain is estimated typically by applying machine learning to magnetic resonance imaging (MRI) data to predict chronological age. The difference between predicted *brain age* and actuackal chronological age (*brain age delta*) reflects the deviation from the expected norm and is often used to index brain health. *Brain age delta* has been related to brain, mental, and cognitive health, and proved valuable in predicting outcomes such as mortality (*Cole et al., 2018*; *Cole and Franke, 2017*; *Elliott et al., 2019*). To different degrees, it is assumed that *brain age delta* reflects past and ongoing neurobiological aging processes (*Cole and Franke, 2017*; *Elliott et al., 2019*; *Franke and Gaser, 2019*; *Smith et al., 2020*). Hence, it is common to interpret positive *brain age deltas* as reflecting a steeper rate of brain aging; often dubbed as accelerated aging (here both terms are used interchangeably) (*Cole and Franke, 2017*; *Franke and Gaser, 2019*; *Smith et al., 2019*).

The assumption that *brain age delta* reflects an ongoing process of faster or slower neurobiological aging implies that there should be a relationship between cross-sectional and longitudinal estimates of *brain age*. Alternatively, individual deviations from the expected *brain age* could capture constant interindividual differences in brain structure that remain stable throughout the lifespan, reflecting early genetic and environmental influences (*Deary, 2012*; *Elliott et al., 2019*; *Walhovd et al., 2016*). These perspectives offer fundamentally divergent interpretations of higher *brain age (delta)* in groups experiencing specific life events, brain disorders, and other medical problems. Here, we tested whether *brain age* – derived from structural T1-weighted (T1w) morphological features – is related to accelerated brain aging, early-life factors, or a combination of both.

If interindividual variations of *brain age* reflect variations in rates of ongoing brain aging (*Figure 1a*), cross-sectional *brain age delta* should be positively associated with brain decline measured longitudinally. Here, we quantified individual brain change as the annual rate of change of *brain age delta* (*brain age delta*$_{long}$). In addition, we also assessed brain change with a composite score of structural brain change as obtained using principal component (PC) analysis of change and change in the different *raw* structural brain features. These analyses were performed in two independent cohorts, both divided into a cross-sectional model generation (training) and a longitudinal, hypothesis testing (test) dataset. If cross-sectional variations in *brain age* reflect differences in brain structure established early in life, one should observe a relationship between *brain age* and influences associated with stable, lifelong effects on brain structure. Here, we selected two congenital factors: self-reported birth weight and polygenic scores for *brain age* (PGS-BA), for which lifelong effects on age-related phenotypes have been shown (*Walhovd et al., 2012*; *Walhovd et al., 2020*; *Figure 1b*). Birth weight reflects normal variation in body (and brain) size as well as prenatal conditions, whereas PGS-BA quantifies genetic liability of having a higher brain age.

**eLife digest** Scientists who study the brain and aging are keen to find an effective way to measure brain health, which could help identify people at risk for dementia or memory problems. One popular marker is 'brain age'. This measurement uses a brain scan to estimate a person's chronological age, then compares the estimated brain age to the person's actual age to determine whether their brain is aging faster or slower than expected for their age.

However, since brain age relies on one brain scan taken at one point in time, it is not clear whether it really measures brain aging or if it might capture brain differences that have been present throughout the individual's life. Studies comparing individual brain scans over several years would be necessary to know for sure.

Now, Vidal-Piñeiro et al. show that the brain-age measurement does not reflect faster brain aging. In the experiments, the researchers compared repeated brain scans of thousands of individuals over 40 years of age. The experiments showed that deviations from normative brain age detected in a single scan reflected early life differences more than changes in the brain over time. For example, people with older-looking brains were more likely to have had a low birth weight or to have a combination of genes associated with having an older looking brain.

Vidal-Piñeiro et al. show that brain age mostly reflects a pre-existing brain condition rather than brain aging. The experiments also suggest that genetics and early brain development likely have a strong impact on brain health throughout life. Future studies trying to test or develop brain-aging measurements should use serial measurements to track brain changes over time.

## Results

### *Brain age* prediction

Chronological age (*Figure 1c*) was predicted based on regional and global features from structural T1w MRI, including cortical thickness, area, volume, and gray-white matter contrast, as well as subcortical volume and intensity imaging-derived phenotypes (|N| = 365). See a list of the different structural features used in the model in *Supplementary files 1 and 2*, and *Figure 1d* for pairwise correlations with age. The model was trained on 38,682 participants (age range = 44.8–82.6 years) with a single MRI from the UK Biobank (*Miller et al., 2016*) using gradient boosting as implemented in XGBoost (https://xgboost.readthedocs.io) and optimized using 10-fold cross-validation and a randomized hyperparameter search. The trained model (*Figure 1e*) was then used to predict *brain age* for an independent test dataset of 1372 participants with two MRIs each (age range = 47.2–80.6 years, mean [SD] follow-up = 2.3 [0.1] years). The predictions – applied to the longitudinal test set – revealed a high correlation between chronological and *brain age* ($r$ = 0.82) with mean absolute error (MAE) = 3.31 years and root mean squared error (RMSE) = 4.14 years (*Figure 1f*), comparable to other *brain age* models using UK Biobank MRI data (*Cole, 2020a*). We used generalized additive models (GAM) to correct for the *brain age* bias, that is, the *underestimation* of brain age in older individuals and vice versa; a regression-to-the-mean bias (*Smith et al., 2019*). *Brain age delta* was calculated as the residual from the GAM fit. *Brain age delta* at baseline and follow-up were strongly correlated ($r$ = 0.81). To establish generalizability, we replicated our results using a different machine learning algorithm – a LASSO-based approach (*Cole, 2020a*) – and an independent training and test (longitudinal) dataset from the Lifebrain consortium (*Walhovd et al., 2018*) with up to 11.2 years of follow-up (3292 unique participants, age range = 18.0–94.4 years; technical and biological replication). See *Figure 1—figure supplement 1* and *Supplementary file 3* for additional demographic information. All the codes used to generate the results are available alongside the article and at https://github.com/LCBC-UiO/VidalPineiro_BrainAge, (*Vidal-Piñeiro, 2021*; copy archived at swh:1:rev:2044c6ca40e0b8f99c9190c6edfde8ca76b559ac).

### *Brain age* does not strongly relate to the rate of brain aging

First, we tested whether cross-sectional *brain age delta* was associated with *brain age delta*<sub>*long*</sub> – that is, annual rate of change in *brain age delta* – using linear models controlling for age, sex, scanning site, and estimated intracranial volume (eICV). We selected the centercept (*brain age delta* at mean chronological age), instead of baseline *brain age delta*, to avoid statistical dependency between

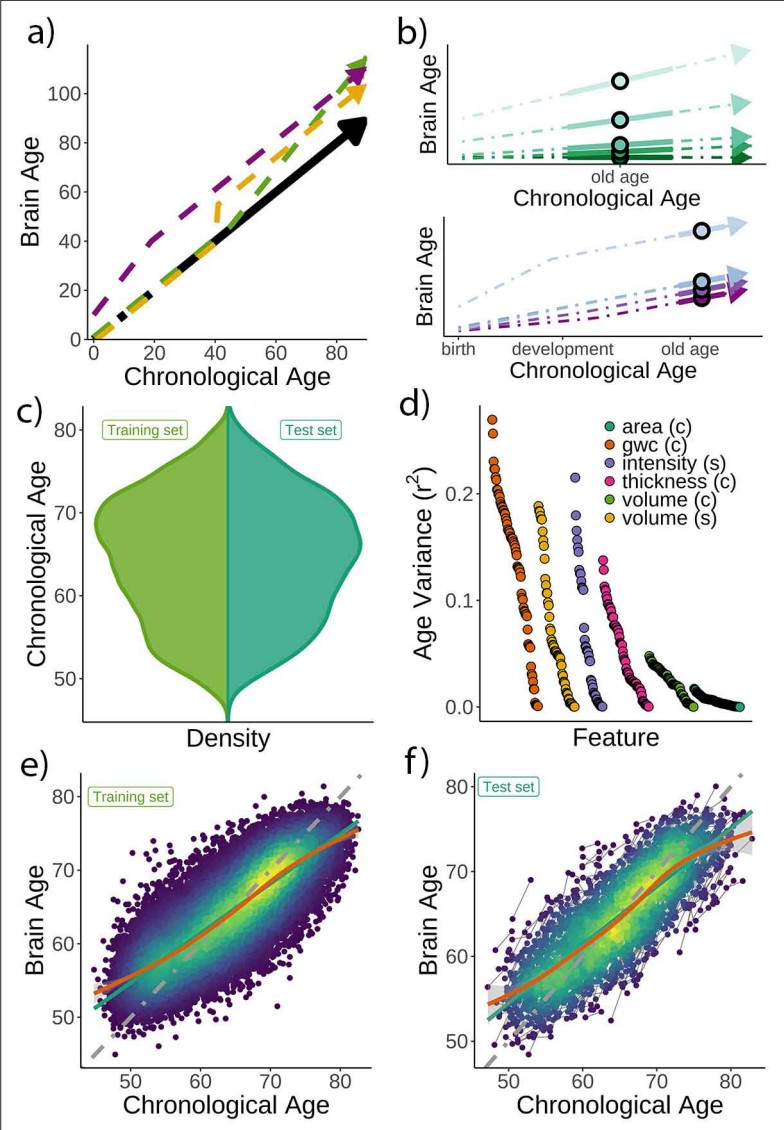

**Figure 1.** Theoretical expectations and study characteristics. (**a**) Three hypothetical trajectories leading to higher *brain age delta*. Higher *brain age delta* can be explained by a steeper rate of neurobiological aging (green), distinct events that led to the accumulation of brain damage in the past (yellow), or early-life genetic and developmental factors (purple). The black arrow represents normative values of *brain age* through the lifespan. (**b**) Brain aging (green) vs. early-life (blue-purple) accounts of *brain age* in older age. For the brain aging notion, cross-sectional *brain age* (points) relates to the slope of *brain age* as assessed by two or more observations across time (continuous line), reflecting ongoing differences in the rate of aging (dashed line, green scale). For the early-life notion, cross-sectional *brain age* (points) relates to early environmental, genetic, and/or developmental differences such as birth weight (blue-purple scale). (**c**) Relative age distribution for the UK Biobank test and training datasets. (**d**) Age variance explained ($r^2$) for each MRI feature in the training dataset. Features are grouped by modality and ordered by the variance explained. (**e**) *Brain age* model as estimated on the training (n = 38,682), and (**f**) test datasets (participants = 1372; two observations each). In (**e**) and (**f**), lines represent the identity (gray; i.e., *f(x) = x or diagonal fit*), the linear (green), and the generalized additive models (GAM; orange) fits of chronological age to *brain age*. Confidence intervals (CIs) around the GAM fit represent 99.9% CIs for the mean. In (**d**), gwc = gray-white matter contrast, (c) = cortical, and (s) = subcortical.

The online version of this article includes the following figure supplement(s) for figure 1:

**Figure supplement 1.** Age distribution for the Lifebrain replication dataset.

**Figure supplement 2.** *Brain age* model predictions.

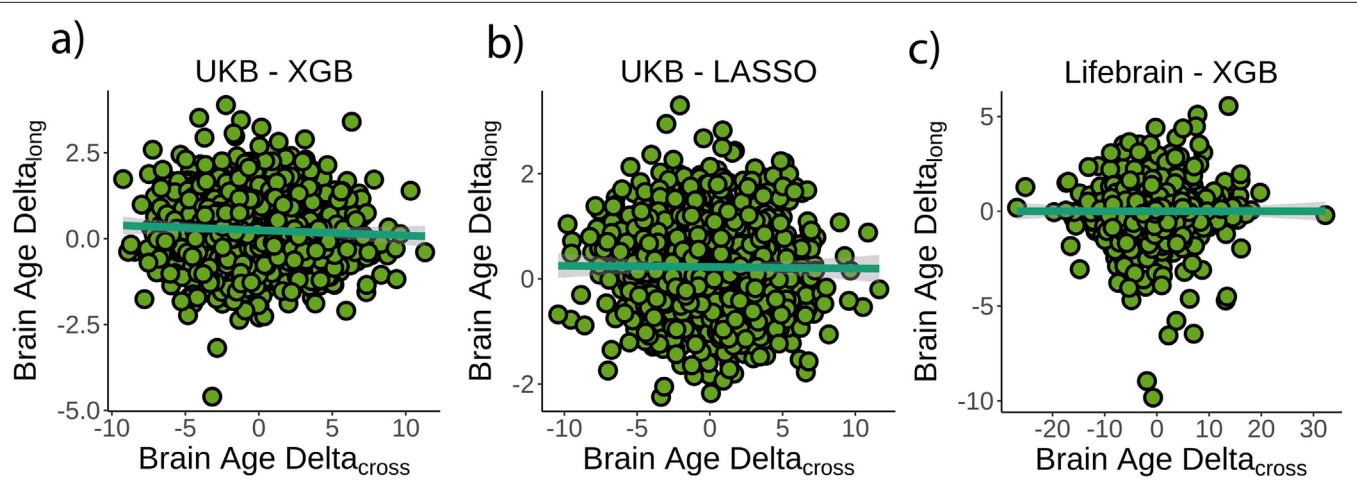

**Figure 2.** Relationship between cross-sectional and longitudinal *brain age delta*. (**a**) Main analysis using the UK Biobank dataset and boosting gradient (n = 1372, p=0.04, $r^2$ = 0.002). (**b**) Replication analyses using a different training algorithm (LASSO; n = 1372, p=0.65, $r^2$ = 0.001) and (**c**) an independent dataset (Lifebrain; n = 1500, p=0.53, $r^2$ = 0.001). XGB = boosting gradient as implemented in XGBoost. Confidence intervals (CIs) represent 99.9% CI for the fit. Longitudinal *brain age delta* (*brain age delta_long*) refers to the rate of change in delta between baseline and follow-up MRI measurements. Cross-sectional *brain age delta* (*brain age delta_cross*) refers to centercept *brain age delta*; that is, at mean age.

The online version of this article includes the following figure supplement(s) for figure 2:

**Figure supplement 1.** Equivalence tests.

**Figure supplement 2.** Relationship between *brain age delta* and composite measures of change.

**Figure supplement 3.** Relationship between *brain age delta* and change in raw features.

indices. Cross-sectional and *brain age delta_long* were weakly, but negatively, associated in the UK Biobank ($\beta$ = –0.016 [±0.008] *delta*/year, t(p) = –2.0 (.04), $r^2$ = 0.002, **Figure 2a**). Cross-sectional and *brain age delta_long* were unrelated using a LASSO regression approach ($\beta$ = –0.003 [±0.006] *delta*/year, t(p) = –0.5 (.65), $r^2$ = 0.001, **Figure 2b**), and in the Lifebrain replication sample ($\beta$ = –0.007 [±0.01] *delta*/year, t(p) = –0.6 (.53), $r^2$ = 0.001, **Figure 2c**). Post-hoc equivalence tests showed that positive relationships with $\beta$ > 0.010 *delta*/year would be rejected in all three analyses, thus confirming a lack of a meaningful relationship between cross-sectional and longitudinal *brain age* (Materials and methods and **Figure 2—figure supplement 1**). UK Biobank (gradient boosting) results remained not significant when *brain age delta* was derived by time points 1 and 2 as two independent training sets (10-fold cross-validation; *uncorrected delta* values), thus avoiding potential confounds with age-bias correction (t(p) = 0.3 (.76)). Lifebrain results remained unaffected after including follow-up interval as an additional covariate or restricting the analysis to participants with long follow-up intervals (>4 years; n = 424). The relationship between cross-sectional and *brain age delta_long* was not significant in both cases ($\beta$ = –0.008 [±0.01] *delta*/year, t(p) = –0.7 (.45); $\beta$ = –0.008 [±0.007] *delta*/year, t(p) = –1.1 (.26)).

We additionally tested whether cross-sectional and longitudinal *brain age delta* (*brain age delta_long*) were associated with a composite measure of longitudinal brain change or with change in any of the structural MRI features. See Materials and methods for details. Cross-sectional *brain age delta* was unrelated to a principal component of change ($\beta$ = –0.009 [±0.01] year, t(p) = –0.7 (.46), $r^2$ = 0.001). We did not find a significant relationship when *brain age delta* was computed with neither a LASSO algorithm nor using the Lifebrain sample ($\beta$ = –0.02 [±0.01] year, t(p) = –1.7 (0.09), $r^2$ = 0.002; $\beta$ = 0.007 [±0.006] year, t(p) = 1.3 (0.2), $r^2$ = 0.001). In contrast, *brain age delta_long* was associated with a principal component of change in the UK Biobank dataset as well as in both replication analyses (all tests p<0.001). See **Figure 2—figure supplement 2** for a visual representation. For specific features, cross-sectional *brain age delta* was significantly related to change – in the expected direction – of features capturing lateral ventricle expansion and white matter hypointensities (p<0.05 Bonferroni-corrected). *Brain age delta_long* related to change in 45 of the features pertaining to four different modalities. The results were replicated both using the LASSO algorithm and the Lifebrain dataset (**Figure 2—figure supplement 3** and **Supplementary file 4**).

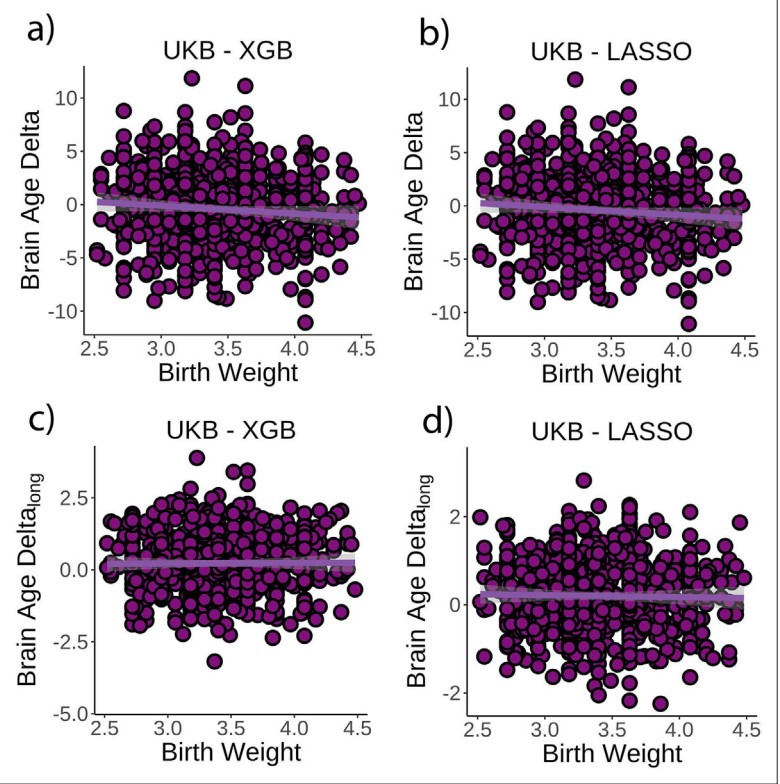

**Figure 3.** Relationship between cross-sectional *brain age delta* and birth weight. (**a**) Main effect of birth weight on *brain age delta* using the UK Biobank dataset and boosting gradient (n = 770, p=0.02, $r^2$ = 0.009). (**b**) This effect was replicated using a different training algorithm (LASSO) (n = 770, p=0.005, $r^2$ = 0.009). Relationship between longitudinal change in *brain age delta* and birth weight was not significant either (**c**) in the main test or (**d**) in the LASSO replication analysis (p>0.5). Note that we used delta at time point 1 to illustrate the main effect of birth weight at time 0 and *brain age delta_long* to represent the birth weight × time interaction of the linear mixed models. Confidence intervals (CIs) represent 99.9% CI for the fit. XGB = boosting gradient as implemented in XGBoost.

The online version of this article includes the following figure supplement(s) for figure 3:

**Figure supplement 1.** Robust effects of birth weight on *brain age delta*.

Finally, we estimated the *rate of aging* effects using a cross-sectional model by estimating the scaling of the size of *delta* with age as defined in *Smith et al., 2019*. The scaling (δ) of *brain age delta* (δ) throughout the datasets' age range was = 0.14 and 0.09 for the UK Biobank and the Lifebrain datasets. This corresponds to an increase in the spread of *brain age delta* of |δ| = 0.38 and 0.37 years – when moving from youngest to oldest – in the UK Biobank and the Lifebrain datasets, suggesting that *brain age delta* only modestly reflects *rate of aging* effects.

### *Brain age delta* is associated with congenital factors on brain structure

Next, we tested whether birth weight was associated with *brain age delta* or change in *brain age delta*. Linear mixed models were used to fit time (from baseline; years), birth weight, and its interaction on *brain age delta* using age at baseline, sex, scanning site, and eICV as covariates. Birth weight was significantly related to *brain age delta* ($\beta$ = –0.70 [±0.30] year/kg, t(p) = –2.3 (0.02), $r^2$ = 0.009, *Figure 3a*), but not to *delta* change ($\beta$ = 0.02 [±0.09] year/kg, t(p) = 0.3 (0.79), *Figure 3c*). Birth weights were limited to normal variations at full term (from 2.5 to 4.5 kg; n = 770 unique individuals) but see *Figure 3—figure supplement 1* for results with varying cutoffs. The results were not affected by excluding individuals being part of multiple births (p=0.02) and were replicated using the LASSO approach ($\beta$ = –0.79 [±0.29] year/kg, t(p) = –2.8 (0.006), $r^2$ = 0.009, *Figure 3b and d*).

Finally, we tested whether PGS-BA related to *brain age delta* and change in *brain age delta* (n = 1339). PGS-BA was computed using a mixture-normal model based on a genome-wide association study (GWAS) of the *brain age delta* phenotype in the UK Biobank training dataset. To test the

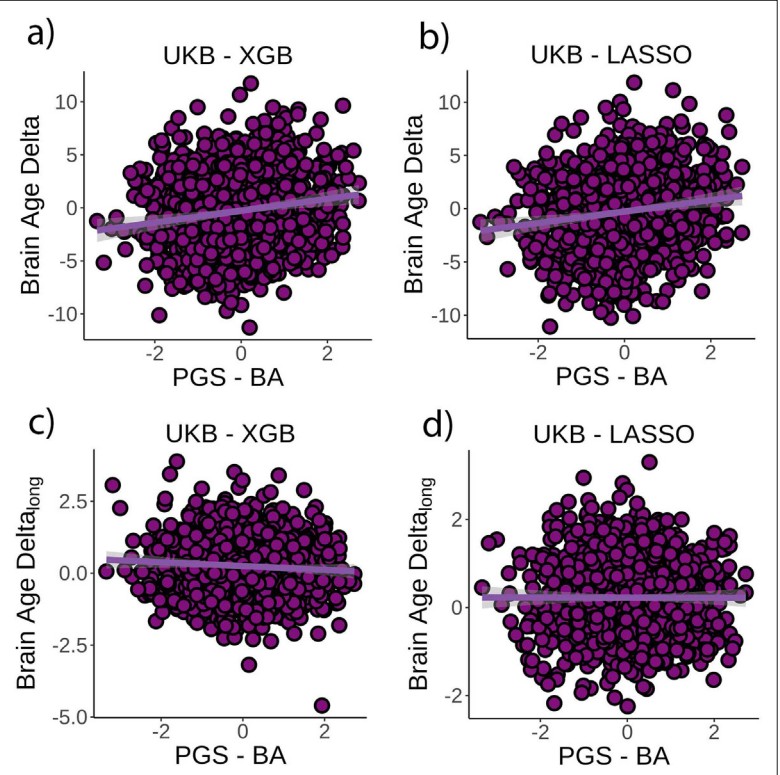

**Figure 4.** Relationship between cross-sectional *brain age delta* and polygenic scores of *brain age delta* (PGS-BA). (**a**) Main effect of PGS-BA on *brain age delta* using the UK Biobank dataset and boosting gradient (n = 1339, p<0.001, $r^2$ = 0.02). (**b**) This effect was replicated using a different training algorithm (LASSO) (n = 1339, p<0.001, $r^2$ = 0.02). (**c**) We found a negative association between longitudinal change in *brain age delta* and PGS-BA (=0.02; higher genetic liability to *brain age* related to negative change in *brain age delta*), which was not found (**d**) in the LASSO replication analysis (p=1.0). Note that we used delta at time point 1 to illustrate the main effect of PGS-BA at time 0 and *brain age delta*$_{long}$ to represent the PGS-BA × time interaction of the linear mixed models. Confidence intervals (CIs) represent 99.9% CI for the fit. XGB = boosting gradient as implemented in XGBoost.

The online version of this article includes the following figure supplement(s) for figure 4:

**Figure supplement 1.** *Brain age delta* genome-wide association study (GWAS).

association, linear mixed models were used as above along with the top 10 genetic PCs to account for population structure. PGS-BA was positively associated with *brain age delta* ($\beta$ = 0.54 [±0.09] year, t(p) = 9.4 (<0.001), $r^2$ = 0.02, *Figure 4a*) and negatively associated with *brain age delta* change ($\beta$ = –0.06 [±0.03] year, t(p) = −2.4 (0.02), *Figure 4c*) in the independent test dataset. Likewise, PGS-BA was associated with *brain age delta* derived from the LASSO algorithm ($\beta$ = 0.53 [±0.09] year, t(p) = 10.4 (<0.001), $r^2$ = 0.02, *Figure 4b*) but not to *brain age delta* change ($\beta$ = –0.001 [±.02] year, t(p) = 0.0 (1.0), *Figure 4d*). See *Figure 4—figure supplement 1* for GWAS results. The association between PGS-BA and *brain age delta* remained significant when using as covariates the top 10 genetic components derived from the full UK Biobank sample (p<0.001 in both analyses).

## Discussion

Altogether, these findings do not support the claim that individual variation in the cross-sectional *brain age* metric capture across-subject differences in the ongoing rate of brain aging. Rather, *brain age* seems to reflect early-life influences on brain structure, and only to a very modest degree reflects actual rate of brain change in middle and old adulthood. A lack of relationship between *brain age* and

rate of brain aging could potentially be explained – although not investigated in the present study – by the effect of circumscribed events such as isolated insults or detrimental lifestyles that occurred in the past, resulting in higher, but not accelerating, *brain age*. Yet, variations in *brain age* could equally reflect congenital and early-life differences and show lifelong stability. Cross-sectional *brain age* studies are ill-suited to disentangle these sources of variation but are often interpreted in line with the former. This assumes that variation in *brain age* largely results from the accumulation of damage and insults during the lifespan, with similar starting points for everyone. An exception is *Elliott et al., 2019*, who found that middle-aged individuals with higher *brain age* already exhibited poorer cognitive function and brain health at age 3 years. This fits a robust corpus of literature showing effects of lifelong, stable influences as indexed by childhood IQ (*Karama et al., 2014*), genetics (*Walhovd et al., 2020*), and neonatal characteristics (*Walhovd et al., 2016*) on brain and cognitive variation in old age.

It has been argued that at a population level *brain age* captures modest *rate of aging* effects because *brain age delta* spreads with increasing age (*Smith et al., 2019*). Here, we found a similar degree of *delta* spreading in our *brain age* metrics. Likewise, our secondary analyses suggested *brain age* related to change in a few specific neuroimaging features, that is, ventricular expansion and white matter hypointensities, though not to any composite score. Thus, both results are compatible and converge towards *brain age* as a real but relatively modest metric for capturing ongoing brain change. The largest part of interindividual variation in *brain age delta*, instead, largely originates before the sample lower bound (≈ 18 and 45 years for the Lifebrain and UK Biobank datasets). Also, associations of *brain age* with other bodily markers of aging or with cognitive decline have yielded mixed support for cross-sectional *brain age* as a marker of individual differences in brain aging (*Cole et al., 2018*, p. 201; *Elliott et al., 2019*; *Franke and Gaser, 2012*). Other multivariate approaches might be better equipped for capturing the dynamics of the aging brain. Using independent component analysis, a recent study found that – compared to a single *brain age* score – distinct modes of multimodal brain variation better reflect both the genetic make-up and ongoing aging effects, with a subset of *modes* showing significant spreading of *delta* with age (*Smith et al., 2020*). The degree to which *brain age* reflects ongoing effects of aging likely depends on the specific features, modalities, and algorithms employed, and is constrained by model properties such as prediction accuracy and homoscedasticity. Yet, without longitudinal imaging, one should not interpret brain age as accelerated aging. Our results align with theoretical claims and empirical observations that covariance structures capturing differences between individuals do not necessarily generalize to covariance structures within individuals (*Molenaar, 2004*; *Schmiedek et al., 2020*). From a measurement theory perspective, our results suggest that cross-sectional *brain age* has low validity as an index of brain aging – despite having high reliability (*Franke and Gaser, 2012*) – as only a small portion of variance is associated with the trait of interest alone (*Zuo et al., 2019*). Most variance is rather associated with other factors that vary systematically across individuals, some of which are already present at birth.

The results further showed that birth weight, which reflects differences in genetic propensities and prenatal environment (*Gielen et al., 2008*), explained a modest portion of the variance in *brain age*. Subtle variations in birth weight are associated with brain structure early in life and present throughout the lifespan (*Walhovd et al., 2016*). This association should be considered as *proof of concept* that the metric of *brain age* reflects the past more than presently ongoing events in the morphological structure of the brain. This was confirmed by the consistent association between PGS-BA and *brain age delta* but not with *brain age delta* change. Since PGS-BA was computed based on cross-sectional *brain age delta*, this relationship may not be surprising, but still suggests a different genetic foundation for longitudinal *brain age*. These findings link with evidence that brain development is strongly influenced by a genetic architecture that, in interaction with environmental factors, leads to substantial, longlasting effects on brain structure. By contrast, aging mechanisms seem to be more related to limitations of maintenance and repair functions and have a more stochastic nature (*Kirkwood, 2005*).

## Limitations and technical considerations

We used large training datasets to estimate the *brain age* models and the PGS scores leading to robust PGS-BA and *brain age* estimates. Self-reported birth weight (*Nilsen et al., 2017*) and cross-sectional *brain age* (*Franke and Gaser, 2012*) are highly reliable measures; thus, our analyses are well-powered to detect small effects (*Zuo et al., 2019*). The reliability of *brain age delta$_{long}$* is, however, unknown. Strictly speaking, *brain age delta* is a prediction error from a model that maximizes the prediction of age in cross-sectional data and thus partially also reflects noise. Given that *delta$_{long}$* is estimated as the difference between two *delta$_{cross}$* estimates, it will hence have higher noise than the cross-sectional estimates, reducing the power in identifying potential associations between longitudinal and cross-sectional delta. This may be compounded by the relatively short interscan interval in the UK Biobank (≈2 years). However, our sample size (n > 1200) ensures that the tests performed in this study are well-powered to detect small effects, even if *delta$_{long}$* has mediocre reliability (*Zuo et al., 2019*). Further, replication of our null results in the Lifebrain sample with more observations and longer follow-up times reduces the likelihood of noise as the main factor behind the lack of relationship. Furthermore, previous studies have found that changes in *brain age* are partly heritable (*Brouwer et al., 2021*) and relate to, for instance, cardiometabolic risk factors (*Beck, 2021*), suggesting that it captures biologically relevant signals (i.e., has predictive validity), although with substantially different origins from cross-sectional *brain age*. Although the reliability of *delta$_{long}$* needs to be formally tested, the null relationship between *delta$_{cross}$* and *delta$_{long}$* does not seem to be a result of a low-powered test.

We speculate that our results partially generalize to other normative and residual-based modeling approaches, as well as to developmental samples. There is considerable evidence in the literature that birth weight and genetic risk for neurodegenerative conditions affect brain structure from early life (*Raznahan et al., 2012*; *Walhovd et al., 2020*; *Walhovd et al., 2016*). *Brain age* models are related to other models such as normative brain charts (*Bethlehem, 2021*; *Dong et al., 2020*) – akin to normative anthropometric charts – the main difference being that *brain age* models predict, rather than control for, age (*Marquand et al., 2019*). Both types of models produce normative brain scores, which are uncorrelated with age (*Butler et al., 2021*). Thus, caution is required when interpreting these scores as indices of brain aging without availability of longitudinal data. Developmental samples may, however, reflect slightly stronger relationships between cross-sectional *brain age delta* and ongoing brain change as brain changes during early-life development typically occur at a faster pace than in middle or later life. Similarly, for specific disease groups such as Alzheimer's disease patients (*Franke and Gaser, 2012*), interindividual brain variation in *brain age* might reflect to a greater extent prevailing loss of brain structure. Moreover, the variance associated with factors other than ongoing development/aging might be more limited in early than later age since influences leading to interindividual variations in brain structure have a shorter span to accumulate. That is, as time from birth increases, chronological age as a marker of individual development is reduced.

Finally, many genetic and environmental factors relate to lifelong stable differences in *brain age* beyond birth weight and PGS-BA. However, both variables are congenital and show stable associations through the lifespan (*Raznahan et al., 2012*; *Walhovd et al., 2020*) without strong evidence that they relate to brain change after adolescence. Thus, birth weight and PGS-BA are paradigmatic for showing how interindividual differences in *brain age* emerge early in life. The present study does not provide a systematic understanding of these influences but presents a framework for interpreting the impact such measures may exert on age-related phenotypes.

## Conclusions

The results call for caution in interpreting brain-derived indices of aging based on cross-sectional MRI data and underscore the need to rely on longitudinal data whenever the goal is to understand the trajectories of brain and cognition in aging.

# Materials and methods

**Key resources table**

| Reagent type (species) or resource | Designation | Source or reference | Identifiers | Additional information |
|---|---|---|---|---|
| Software, algorithm | R Project for Statistical Computing | https://www.r-project.org/ | RRID:SCR_001905 | Version 3.6.3 |
| Software, algorithm | FreeSurfer | https://surfer.nmr.mgh.harvard.edu/ | RRID:SCR_001847 | Version 6.0 |

## Participants and samples

The main sample was drawn from the UK Biobank neuroimaging branch (https://www.ukbiobank.ac.uk/ *Miller et al., 2016*). 38,682 individuals had MRI available at a single time point and were used as the training dataset. 1372 individuals had longitudinal data and were used as the test dataset. The present analyses were conducted under data application number 32048. The Lifebrain dataset (*Walhovd et al., 2018*) included datasets from five different major European Lifespan cohorts: the Center for Lifespan Changes in Brain and Cognition cohort (LCBC, Oslo; *Walhovd et al., 2016*), the Cambridge Center for Aging and Neuroscience study (Cam-CAN; *Shafto et al., 2014*; *Taylor et al., 2017*), the Berlin Study of Aging-II (Base-II; *Bertram et al., 2014*), the University of Barcelona cohort (UB; *Rajaram et al., 2016*; *Vidal-Piñeiro et al., 2014*), and the BETULA project (Umeå; *Nilsson et al., 2010*). Furthermore, we included data from the Australian Imaging Biomarkers and Lifestyle flagship study of ageing (AIBL; *Ellis et al., 2009*). In addition to cohort-specific inclusion and exclusion criteria, individuals aged <18 years, or with evidence of mild cognitive impairment, or Alzheimer's disease were excluded from the analyses. 1792 individuals with only one available scan were used for the Lifebrain training dataset. 1500 individuals with available follow-up of >0.4 years were included in the test dataset. Individuals had between 2 and 8 available scans each. Sample demographics for the UK Biobank and the Lifebrain samples are provided in *Supplementary file 3*. See also *Figure 1c* and *Figure 1—figure supplement 1* for a visual representation of the age distribution in the UK Biobank and the Lifebrain datasets. UK Biobank (North West Multi-Center Research Ethics Committee [MREC]; see also https://www.ukbiobank.ac.uk/the-ethics-and-governance-council) and the different cohorts of the Lifebrain replication dataset (*Supplementary file 5*) have ethical approval from the respective regional ethics committees. All participants provided informed consent.

## MRI acquisition and preprocessing

See https://biobank.ctsu.ox.ac.uk/crystal/crystal/docs/brain_mri.pdf for details on the UK Biobank T1w MRI acquisition. UK Biobank and Lifebrain MRI data were acquired with 3 and 10 different scanners, respectively. T1w MRI acquisition parameters for both the Lifebrain and the UK Biobank are summarized in *Supplementary file 6*.

We used summary regional and global metrics derived from T1w data. For UK Biobank, we used the imaging-derived phenotypes developed centrally by UK Biobank researchers (*Miller et al., 2016*) and distributed via the data showcase (http://biobank.ctsu.ox.ac.uk/crystal/index.cgi). See preprocessing details in https://biobank.ctsu.ox.ac.uk/crystal/crystal/docs/brain_mri.pdf. This procedure yielded 365 structural MRI features, partitioned in 68 features of cortical thickness, area, and gray-white matter contrast, 66 features of cortical volume, 41 features of subcortical intensity, and 54 features of subcortical volume. See the list of features in *Supplementary files 1 and 2*. Lifebrain data were processed on the Colossus processing cluster, University of Oslo. Similar to the UK Biobank pipeline, we used the fully automated longitudinal FreeSurfer v.6.0. pipeline (*Reuter et al., 2012*) for cortical reconstruction and subcortical segmentation of the structural T1w data (http://surfer.nmr.mgh.harvard.edu/fswiki *Dale et al., 1999*; *Fischl et al., 1999*; *Fischl and Dale, 2000*) and used similar atlases for structural segmentation and feature extraction.

## Birth weight

We used birth weight (kg) from the UK Biobank (*field #20022*). Participants were asked to enter their birth weight at the initial assessment visit, the first repeat assessment visit, or the first imaging visit. In the case of multiple birth weight instances, we used the latest available input. n = 894 participants from the test dataset had available data on birth weight. The main analysis was constrained to normal variations in birth weight between 2.5 and 4.5 kg (n = 770; *Walhovd et al., 2012*) due to lower

reliability of extreme scores and to tentatively remove participants potentially with severe medical complications associated with prematurity.

## Genetic preprocessing

Detailed information on genotyping, imputation, and quality control was published by *Bycroft et al., 2018*. For genetic analyses, we only included participants with both genotypes and MRI scans. Following the recommendations from the UK Biobank website, we excluded individuals with failed genotyping, who had abnormal heterozygosity status, or who withdrew their consents. We also removed participants who were genetically related – up to the third degree – to at least another participant as estimated by the kinship coefficients as implemented in PLINK (*Chang et al., 2015*). For the GWAS we used 38,163 individuals from the training dataset. Polygenic risk scores were computed using the test dataset consisting of 1339 individuals with longitudinal MRI.

## Genome-wide association study (GWAS)

We performed GWAS analysis on the training dataset and the *brain age delta-semi*-corrected phenotype using the imputed UK Biobank genotypes. To control for subtle effects of population stratification in the dataset, we computed the top 10 PCs using the PLINK command *–pca* on a decorrelated set of autosome single-nucleotide polymorphisms (SNPs). The set of SNPs (n = 101,797) were generated by using the PLINK command, `--maf 0.05`, `--hwe 1e`$^{-6}$, `--indep-pairwise 100 50 0.1`. The *–glm* function from PLINK was used to perform GWAS on about 9 million autosomal SNPs, including age, sex, and the top 10 PCs as covariates. See Manhattan and quantile-quantile (QQ) plots in *Figure 4—figure supplement 1*. Note that our results corroborated the same association region reported in *Jonsson et al., 2019* with a smaller sample.

## Polygenic scores (PGS)

The GWAS results for the training dataset were used to compute PGS (PGS-BA) in the independent test dataset (n = 1339 participants). We used the recently developed method PRS-CS (*Ge et al., 2019*) to estimate the posterior effect sizes of SNPs that were shown to have high quality in the HapMap data (*International HapMap 3 Consortium et al., 2010*). Rather than estimating the polygenicity of *brain age delta* from our data, we assumed a highly polygenic architecture for *brain age delta* by setting the parameter *--phi = 0.01* (*Boyle et al., 2017*). The remaining parameters of PRS-CS were set to the default values. PGS was based on 654,725 SNPs and was computed on the independent test data using the *--score* function from PLINK. SNPs were aligned with HapMap 3 SNPs (autosome only as provided by PRC-CS) and posterior effects were estimated. We also computed the population structures PCs' in the test dataset using the same procedure as in the training dataset.

## Statistical analyses

All statistical analyses were run with R version 3.6.3 https://www.r-project.org/. We used the UK Biobank as the main sample and the Lifebrain cohort for independent replication. The main description refers to the UK Biobank pipeline, though Lifebrain replication followed identical steps unless otherwise stated. For replication across machine learning pipelines, we used a LASSO regression approach for age prediction, adapted from (*Cole, 2020b*). See more details in *Cole, 2020a*. The correlation between LASSO-based and Gradient Boosting-based *brain age deltas* was 0.80.

## *Brain age* prediction

We used machine learning to estimate each individuals' *brain age* based on a set of regional and global features extracted from T1w sequences. We estimated *brain age* using gradient tree boosting (https://xgboost.readthedocs.io). We used participants with only one MRI scan for the training dataset (n = 36,682) and participants with longitudinal data as test dataset (n = 1372). All variables were scaled prior to any analyses using the training dataset metrics as reference.

The model was optimized in the training set using a 10-fold cross-validation randomized hyperparameter search (50 iterations). The hyperparameters explored were number of estimators [seq(100:600, by = 50)], learning rate (0.01, 0.05, 0.1, 0.15, 0.2), maximum depth [seq(2:8, by = 1)], gamma regularization parameter [seq(0.5:1.5, by = 0.5)], and min child weight [seq(1:4, by = 1)]. The remaining parameters were left to default. The optimal parameters were number of estimators = 500, learning

rate = 0.1, maximum depth = 5, gamma = 1, and min child weight = 4 predicting $r^2$ = 0.68 variance in chronological age with MAE = 3.41 and RMSE = 4.29. See visual representation in *Figure 1f*.

Next, we recomputed the machine learning model using the entire training dataset and the optimal hyperparameters and used it to predict *brain age* for the test dataset (*Figure 1e*). These metrics are similar or better than other *brain age* models using UK Biobank MRI data (*Cole, 2020a*; *de Lange et al., 2019*) and the cross-validation diagnostics. We used GAM to correct for the *brain age* bias estimation (*Smith et al., 2019*); $r$ = –0.54 for the test dataset. Note that we used GAM fittings as estimated in the training dataset so *delta* values in the test dataset are not centered to 0. *Brain age delta* was estimated as the GAM residual. The correlation between *brain age delta* corrected based on the training vs. the test fit was $r$ > 0.99. Also, GAM-based bias correction led to similar *brain age delta* estimations to linear and quadratic-based corrections ($r$ > 0.99). The diagnostics for the LASSO-based model were as follows: variance explained ($r^2$) = 0.69/0.69; MAE = 3.36/3.28; RMSE = 4.21/4.04; age bias = –0.56/–0.52 for the training and predicted datasets. See representation of the *brain age* prediction in *Figure 2—figure supplement 2*.

## Higher level analysis

### Relationship between cross-sectional and longitudinal *brain age*

For each participant, we computed the mean *brain age delta* across the two MRI time points and the yearly rate of change (*brain age delta$_{long}$*). We selected mean, instead of baseline *brain age delta*, to avoid statistical dependency between both indices (*Rogosa and Willett, 1985*; *Wainer, 2000*). *Brain age delta$_{long}$* was fitted by mean *brain age delta* using a linear regression model, which accounted for age, sex, site, and eICV. We used mean eICV across both time points.

### Relationship between *brain age delta* and change in brain features

For each participant, we computed the yearly rate of change in all the *raw* neuroimaging features and tested whether change was significantly different from 0 (one-sample t-test, <0.05, Bonferroni-corrected; *Figure 2—figure supplement 3*, *Supplementary file 4*). Features with significant change over time were fed into a PC analysis (uncentered). The first component, explaining ≃20% of the variance both in the UK Biobank and the Lifebrain datasets, was selected for further analysis. Although it did not qualitatively affect the results, we removed two and three extreme outliers from the UK Biobank and Lifebrain datasets (score >10). See *Supplementary file 4* for component weights. Finally, we tested whether cross-sectional and *brain age delta$_{long}$* predicted brain change as quantified both by the first component analysis and change in each of the *raw* neuroimaging features (p<0.05, Bonferroni-corrected) using the same models described above.

### Spreading of *brain age delta* with age

Further, we estimated the degree to which *brain age delta* reflects *rate of aging* using a cross-sectional model proposed by *Smith et al., 2019*, which estimates the scaling of *brain age delta* through the datasets' age range. The scaling is estimated by $\lambda$ in $\delta = \delta_0(1 + \lambda Y_0)$, where $\delta$ is *brain age delta*, $Y_0$ is a linear mapping of chronological age into the range 0:1, and $|\delta_0|$ relates to *brain age delta* distribution in the youngest participants. The *spread* of *brain age delta* throughout the datasets' age range can then be expressed as $|\delta_0| \lambda$ (years).

### Relationship between *brain age* PGS and cross-sectional and longitudinal *brain age*

This association was tested using linear mixed models with time from baseline (years), PGS-BA, and its interaction on *brain age delta*. Age at baseline, sex, site, eICV, and the 10 first PCs for population structure were used as covariates. The PCs of population structure were added to minimize false positives associated with any form of relatedness within the sample.

### Effects of birth weight on *brain age*

Linear mixed models were used to fit time, birth weight, and its interaction on *brain age delta*, using age at baseline, sex, site, and eICV as covariates. We explored the consistency of the results by modifying the birth weight limits in a grid-like fashion [0.5, 2.7, 0.025] and [4.2, 6.5, 0.025] for minimum

and maximum birth weight (*Figure 3—figure supplement 1*). Self-reported birth weight is a reliable estimate of actual birth weight. However, extreme values are either misestimated or reflect profound gestational abnormalities (*Nilsen et al., 2017*; *Tehranifar et al., 2009*).

### Equivalence tests

Post-hoc equivalence tests were performed to test for the absence of a relationship between cross-sectional and *brain age delta*$_{long}$ (*Lakens et al., 2018*). Specifically, we used inferiority tests to test whether a null hypothesis of an effect at least as large as Δ (in years/*delta*) could be rejected. We reran the three main models assessing a relationship between cross-sectional and longitudinal *brain age delta* (UK Biobank trained with boosting gradient, UK Biobank trained with LASSO, and Lifebrain trained with boosting gradient) varying the right-hand-side test (Δ) [–0.02, 0.05, 0.001] ($p < 0.05$, one-tailed; *Figure 2—figure supplement 1*).

Assumptions were checked for the main statistical tests using plot diagnostics. Variance explained for single terms refers to unique variance (UVE), which is defined as the difference in explained variance between the full model and the model without the term of interest. For linear mixed models, UVE was estimated as implemented in the *MuMIn* r-package.

## Lifebrain-specific steps

### Features

The Lifebrain cohort included |N| = 372 features. It included eight new features compared to the UK Biobank dataset, whereas one feature was excluded (new features: left and right temporal pole area volume and thickness, cerebral white matter volume, cortex volume; excluded feature: ventricle choroid). See age variance explained in each feature in *Supplementary files 1 and 2* as estimated with GAMs.

### Quality control

Prior to any analysis, we tentatively removed observations for which > 5% of the features fell above or below 5 SD from the sample mean. The application of this arbitrary high threshold led to the removal of 10 observations. We considered these MRI data to be extreme outliers and likely to be artifactual and/or contaminated by important sources of noise. Also, before brain prediction, we tentatively removed variance associated with the different scanners using generalized additive mixed models (GAMM) and controlling for age as a smooth factor and a subject identifier as random intercept. This correction was performed due to differences in age distribution by scanner and lack of across scanner calibration.

### Hyperparameter search and model diagnostics

The optimal parameters for the Lifebrain replication sample were number of estimators = 600, learning rate = 0.05, maximum depth = 4, gamma = 1.5, and min child weight = 1. Using cross-validation, the model predicted $r^2$ = 0.92 of the age variance with MAE = 4.75 and RMSE = 6.31. *Brain age* was underestimated in older age (bias $r$ = –0.33).

### Model prediction

The age variance explained by *brain age* was $r^2$ = 0.90 with MAE = 4.68 and RMSE = 6.06. *Brain age* was underestimated in older age (bias $r$ = –0.25; *Figure 1—figure supplement 2*).

### Higher level analysis

For each individual, mean *brain age delta* was considered as the grand mean *brain age delta* across the different MRI time points. To compute *brain age delta*$_{long}$ , we set for each participant a linear regression model with observations equal to the number of time points that fitted *brain age delta* by time since the initial visit. Slope indexed change in *brain age delta*/year. The relationship between mean and *brain age delta*$_{long}$ was tested using linear mixed models controlling for age, sex, and eICV as fixed effects, and using a site identifier as a random intercept. Likewise, linear mixed models were used to test the relationship between *brain age delta* and change in brain features. Note that eICV was identical across time points as a result of being estimated through the longitudinal FreeSurfer

pipeline. We could not obtain the required information on genetics and birth weight to replicate the analyses supporting the *early-life* account.

## Data and code availability

The raw data were gathered from the UK Biobank, the Lifebrain cohort, and the AIBL. Raw data requests are specific to each cohort. UK Biobank and AIBL data are available upon application to UK Biobank and at https://aibl.csiro.au upon corresponding approvals. For the Lifebrain cohorts, requests for raw MRI data should be submitted to the corresponding principal investigator. See contact details in *Supplementary file 5*. MRI data is not openly available as participants did not consent to share publicly their data. Access to data is available upon reasonable requests and transfer agreements. Different sample agreements are required for each dataset.Statistical analyses in this article are available alongside the article and will be available at https://github.com/LCBC-UiO/VidalPineiro_BrainAge. All analyses were performed in R 3.6.3. The scripts were run on the Colossus processing cluster, University of Oslo. UK Biobanks' data acquisition, MRI preprocessing, and feature generation pipelines are freely available (https://www.fmrib.ox.ac.uk/ukbiobank). For the Lifebrain cohorts, the image acquisition details are summarized in *Supplementary file 6*. MRI preprocessing and feature generation scripts were performed with the freely available FreeSurfer software (https://surfer.nmr.mgh.harvard.edu/). For bash-sourcing scripts, please contact the corresponding author.

## Acknowledgements

BASE-II has been supported by the German Federal Ministry of Education and Research under grant numbers 16SV5537/16SV5837/16SV5538/16SV5536K/01UW0808/01U-W0706/01GL1716A/01GL1716B. Part of the computation was performed on the Norwegian high-performance computation resources, sigma2, through the project no. nn9769k. The Wellcome Centre for Integrative Neuroimaging is supported by core funding from award 203139/Z/16/Z from the Wellcome Trust. Data used in the preparation of this article were partially obtained from the AIBL funded by the Commonwealth Scientific and Industrial Research Organisation (CSIRO), which was made available at the ADNI database (http://www.loni.usc.edu/ADNI). UK Biobank is generously supported by its founding funders the Wellcome Trust and UK Medical Research Council, as well as the Department of Health, Scottish Government, the Northwest Regional Development Agency, British Heart Foundation and Cancer Research UK. The organization has over 150 dedicated members of staff, based in multiple locations across the UK.

## Additional information

### Competing interests

Christian A Drevon: CAD: Is an employee of Vitas Ltd. The other authors declare that no competing interests exist.

### Funding

| Funder | Grant reference number | Author |
| --- | --- | --- |
| H2020 European Research Council | 732592 | Kristine Beate Walhovd |
| H2020 European Research Council | 283634 725025 | Anders Fjell |
| H2020 European Research Council | 313440 | Kristine Beate Walhovd |
| Norges Forskningsråd | | Anders Fjell |
| Max Planck Institute for Dynamics of Complex Technical Systems Magdeburg | | Andreas M Brandmaier |

| Funder | Grant reference number | Author |
|---|---|---|
| María de Maeztu Unit of Excellence (Institute of Neurosciences,University of Barcelona) | MDM-2017-0729 | Barbara Segura Carme Junqué David Bartres-Faz |
| European Research Council | 677804 | Simone Kühn |
| UK Medical Research Council | G1001354 | Sana Suri Enikő Zsoldos |
| Charitable Trust | 1117747 | Enikő Zsoldos Sana Suri |
| Alzheimer's Research UK | 441 | Sana Suri |
| NIHR Biomedical Research Centre, Oxford | | Sana Suri |
| Knut and Alice Wallenberg Foundation | | Lars Nyberg |
| ICREA Academia Award | | David Bartres-Faz |
| Norges Forskningsråd | 324882 | Didac Vidal-Pineiro |
| Medical Research Council | SUAG/046 G101400 | Richard N Henson |

The funders had no role in study design, data collection and interpretation, or the decision to submit the work for publication.

## Author contributions

Didac Vidal-Pineiro, Conceptualization, Formal analysis, Visualization, Writing – original draft, Writing – review and editing; Yunpeng Wang, Conceptualization, Formal analysis, Visualization, Writing – review and editing; Stine K Krogsrud, Sandra Düzel, Barbara Segura, Rene Westerhausen, Enikő Zsoldos, Data curation, Writing – review and editing; Inge K Amlien, Data curation, Resources, Software; William FC Baaré, David Bartres-Faz, Lars Bertram, Richard N Henson, Carme Junqué, Rogier Andrew Kievit, Simone Kühn, Kathrine S Madsen, Sana Suri, Data curation, Funding acquisition, Writing – review and editing; Andreas M Brandmaier, Funding acquisition, Methodology, Writing – review and editing; Christian A Drevon, Data curation, Resources, Writing – review and editing; Klaus Ebmeier, Funding acquisition, Writing – review and editing; Esten Leonardsen, Øystein Sørensen, Formal analysis, Methodology, Writing – review and editing; Ulman Lindenberger, Lars Nyberg, Conceptualization, Data curation, Funding acquisition, Writing – review and editing; Fredrik Magnussen, Data curation, Methodology, Resources, Writing – review and editing; Athanasia Monika Mowinckel, Resources, Visualization, Writing – review and editing; James M Roe, Formal analysis, Writing – review and editing; Stephen M Smith, Conceptualization, Methodology, Writing – review and editing; Andrew Zalesky, Conceptualization, Writing – review and editing; Kristine Beate Walhovd, Conceptualization, Funding acquisition, Supervision, Writing – review and editing; Anders Fjell, Conceptualization, Funding acquisition, Supervision, Writing – original draft, Writing – review and editing

## Author ORCIDs

Didac Vidal-Pineiro http://orcid.org/0000-0001-9997-9156
David Bartres-Faz http://orcid.org/0000-0001-6020-4118
Andreas M Brandmaier http://orcid.org/0000-0001-8765-6982
Christian A Drevon http://orcid.org/0000-0002-7216-2784
Klaus Ebmeier http://orcid.org/0000-0002-5190-7038
Rogier Andrew Kievit http://orcid.org/0000-0003-0700-4568
Simone Kühn http://orcid.org/0000-0001-6823-7969
Ulman Lindenberger http://orcid.org/0000-0001-8428-6453
Fredrik Magnussen http://orcid.org/0000-0003-2574-1705
Athanasia Monika Mowinckel http://orcid.org/0000-0002-5756-0223
Lars Nyberg http://orcid.org/0000-0002-3367-1746
Stephen M Smith http://orcid.org/0000-0001-8166-069X
Enikő Zsoldos http://orcid.org/0000-0002-0478-6165

### Ethics

UK Biobank (North West Multi-Center Research Ethics Committee [MREC]; see also https://www.ukbiobank.ac.uk/the-ethics-and-governance-council) and the different cohorts of the Lifebrain replication dataset (see Pseudo-Table below) have ethical approval from the respective regional ethics committees. All participants provided informed consent. LCBC Norwegian Regional Committee for Medical and Health Research Ethic, Regional Ethical Committee of South Norway, BETULA Regional Ethical Vetting Board at Umeå University, BASE-II Ethics committee of the Charité-Universitätsmedizin Berlin Cam-CAN, Cambridgeshire 2 Research Ethics Committee, UB Comisión de Bioética de la Universidad de Barcelona and Hospital Clinic AIBL Institutional ethics committees of Austin Health, StVincent's Health Hollywood Private Hospital and Edith Cowan University.

### Decision letter and Author response

Decision letter https://doi.org/10.7554/eLife.69995.sa1
Author response https://doi.org/10.7554/eLife.69995.sa2

---

## Additional files

### Supplementary files

• Supplementary file 1. List of cortical brain features. List of cortical features included in the *brain age* model and age variance explained in the UK Biobank and the Lifebrain training datasets. Vol = volume; GWC = gray-white matter contrast; Cth = cortical thickness.

• Supplementary file 2. List of subcortical brain features. List of subcortical features included in the *brain age* model and age variance explained in the UK Biobank and the Lifebrain training datasets. Vol = volume; Int = intensity; hemi = hemisphere.

• Supplementary file 3. Sociodemographics. Main sample descriptives for the training and test datasets. Obs = mean number of observations per participant (SD). Follow-up = mean time (years) between the first and the last MRI observation (SD). For the test datasets, age and age range refer to age at baseline. *AIBL does not belong to the Lifebrain consortium but was included to enrich the replication sample.

• Supplementary file 4. Relationship between *brain age delta* and change in brain features. Long. change = longitudinal change in the *raw* neuroimaging features (mean change [$\log_{10}(p)$]). PC1 load = feature loadings on the first component of longitudinal change. $Delta_{cross}$ = relationship between cross-sectional *brain age delta* and feature change ($r^2$ [$\log_{10}(p)$]). $Delta_{long}$ = relationship between longitudinal *brain age delta* and feature change ($r^2$ [$\log_{10}(p)$]). GWC = gray-white matter contrast. Cth = cortical thickness. Bil = bilateral. Subc = subcortical. n = 1372 and 1500 for the UK Biobank and the Lifebrain datasets. |N| = 365 and 372 features in the UK Biobank and the Lifebrain datasets. XGB = boosting gradient as implemented in XGBoost.

• Supplementary file 5. Contact information. Contact information and ethical comittees for the different cohorts.

• Supplementary file 6. Data acquisition parameters. Data acquisition parameters for the T1w sequences. *UK Biobank employed three scanners of the same model and with equivalent parameters (Cheadle, Reading, and Newcastle centers). **AIBL does not belong to the Lifebrain consortium but was included in the Lifebrain replication dataset.

• Transparent reporting form

• Source code 1. Analysis Code.

### Data availability

The raw data were gathered from the UK Biobank, the Lifebrain cohort, and the AIBL. Raw data requests are specific to each cohort. UK Biobank and AIBL data are available upon application to UK Biobank and at https://aibl.csiro.au upon corresponding approvals. For the Lifebrain cohorts, requests for raw MRI data should be submitted to the corresponding principal investigator. See contact details in Supplementary File 5. Different agreements are required for each dataset. Statistical analyses in this manuscript are available alongside the manuscript and will be made available at https://github.com/LCBC-UiO/VidalPineiro_BrainAge, (copy archived at swh:1:rev:2044c6ca40e0b8f99c9190c6edfde-8ca76b559ac). All analyses were performed in R 3.6.3. The scripts were run on the Colossus processing cluster, University of Oslo. UK Biobanks' data acquisition, MRI preprocessing, and feature generation

pipelines are freely available (https://www.fmrib.ox.ac.uk/ukbiobank). For the Lifebrain cohorts, the image acquisition details are summarized in Supplementary File 6. MRI preprocessing and feature generation scripts were performed with the freely available FreeSurfer software (https://surfer.nmr. mgh.harvard.edu/).

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
