## [Decision Letter]

**Acceptance summary:**

This paper is of interest to scientists within the field of lifespan developmental neuroscience. It revealed that BrainAge scores did not predict within-person aging of the brain taken from the longitudinal data, but did correlate with metrics already occurring at birth, namely birth weight and polygenetic scores. This calls cautions in interpreting cross-sectional BrainAge indices as well as concluding their validity as markers of individual-level brain aging process.

**Decision letter after peer review:**

Thank you for submitting your article "Individual variations in "Brain age" relate to early life factors more than to longitudinal brain change" for consideration by *eLife*. Your article has been reviewed by 2 peer reviewers, and the evaluation has been overseen by a Reviewing Editor and Christian Büchel as the Senior Editor. The following individual involved in review of your submission has agreed to reveal their identity: Xi-Nian Zuo (Reviewer #2).

Essential Revisions:

1) Improve terminology precision and interpretation (e.g. early life influence).

2) Improve manuscript readability significantly (including figures).

3) Add study limitations and future work (including developing brain).

4) Introduce measurement theory of individual differences.

5) Clarify the relationship between brainage and other metrics.

6) Discuss/clarify the effect size and the possible implications.

*Reviewer #1 (Recommendations for the authors):*

Beyond what was summarized in the Public Review, here I note some errors in figures and text so that the authors can fix those, and suggestions for edits that should help with clarity.

As already mentioned, the purpose of this study is to be applauded as mis- and over-interpretation of cross-sectional data findings as generalizing to true longitudinal change within individuals is a major problem in our field. The increasingly popular use of BrainAge as a metric has only increased this problem. The authors attempt to tests these assumptions was much needed.

The format of the submission made the manuscript very difficult to parse (and I work immediately in this exact field). Primarily driving this difficult to understand the details of the paper are (1) the extreme brevity of the background/introduction and (2) the unfortunate reverse order of the manuscript sections -- it is always difficult when the results come before the methods or description of variables, but in this particular case, it was extremely disadvantageous. If this is the required journal format, then I guess not much van be done, but it is a hindrance to understanding the study. I found I could not read the Results section on its own and had to flip through to simultaneously read the Methods and Supplements. This was exhausting. Some readers may simply give up. One suggestion (highly recommended) is to utilize the Abstract to at least introduce some specifics about the study, like the variables to be used (morphological structural features instead of brain integrity or brain decline – no one can tell what the study will use as written). Similarly I did not know what polygenic scores were based on or what genetic amalgamation they represented until very far into the paper. Also portions of the intro were not clear on the first read, and only made sense after I finished the whole paper and supplemental info. Some word changes may help?

Discussion: I thought that calling birth weight and genetics "early-life influences" throughout the paper a bit of an overstatement. Because this calls to mind many environmental things one experiences as a developing child/adolescent and was misleading (esp since things are not defined until very late in the sections, like Method at the end). A more accurate thing to call this would be something that made it clear that these variables represented "at birth indices". I do think this is a VERY important distinction and keeps the authors from leading the readers into a concept that is vague and perhaps not what is expected from that terminology.

Some study limitations are always warranted and a thoughtful inclusion. There really aren't any provided here to speak of. As a small example, I wondered what the impact is of very different sample sizes in different tests, but then comparisons across those analyses being made (as in the 770 for the birth weight, vs the 38k and 1372 for the other tests).

*Reviewer #2 (Recommendations for the authors):*

1. Measurement theory of individual differences should be included somewhere to clarify different focuses in terms of its two components: inter-individual (between-subject) variability and intra-individual (within-subject) variability. In theory, cross-sectional data is not enough to separate the two components and thus not good enough for any studies of individual differences, which call longitudinal data in ideal experimental design, to ensure both reliability and validity. Please see some comments in a recent publication (https://www.nature.com/articles/s41562-019-0655-x).

2. BrainAge is a recently developed method for normative aging research. What is the relationship between BrainAge and those metrics derived from the growth charts (e.g., height and weight).

3. I noticed that the effect sizes detected are all small (e.g., the R2 < 0.01). In such large sample, how these weak associations are interpreted in terms of the measurement reliability and validity? What is the potential factors with impacts on the ability of detecting such small effect size?

4. This work is demonstrated for aging samples, but should be generalizable for developing samples. Please discuss about this point more comprehensively. Some recent work (e.g., https://www.sciencedirect.com/science/article/pii/S2095927320304965) may be of values.

---

## [Author Response]

Essential Revisions:1) Improve terminology precision and interpretation (e.g. early life influence).

Clarification of terminology is discussed in our response to Reviewer #1, comment #8. Briefly, in the revised version of the manuscript, the term “early-life influences” is replaced with “congenital factors” when referring to birth weight and polygenic scores. Further, in the revised version, we discuss in the limitation sections to which extent the relationship between cross-sectional brain age and the congenital factors can be extrapolated to other congenital and early-life variables.

2) Improve manuscript readability significantly (including figures).

We have carefully gone through the whole manuscript and revised the text to improve readability. This is explained in detail in the response to Reviewer #1, comment #1. To ease readability, the manuscript now includes additional details regarding the study both in the abstract and in the introduction. We also followed the reviewer’s suggestion of including additional plots for clarity and transparency and we have modified the existing plots following his/her suggestions.

3) Add study limitations and future work (including developing brain).

The limitations of the study are discussed in depth in response to Reviewer #1, comment #11. The revised manuscript now includes a “Limitations” section in which we discuss:

1) Statistical power;

2) The extent to which our results can be generalized to other normative modeling approaches and developmental samples, and

3) Whether the relationship between brain age and the congenital factors polygenic scores for brain age (PGS-BA) and birth weight can be extended to other congenital and early life factors.

4) Introduce measurement theory of individual differences.

This issue is discussed in response to Reviewer #2, comment #1. The relationship between cross-sectional and longitudinal brain age can be understood in the context of the measurement theory of individual differences. Following this account, the results indicate cross-sectional brain age has low validity – despite high reliability – rather reflecting variance related to factors that vary systematically across individuals and that are present early in life.

5) Clarify the relationship between brainage and other metrics.

This issue is discussed in response to Reviewer #2, comment #2. Normative brain aging charts are analogous to normative anthropometric growth charts used in pediatrics. Brain age models can be considered a special case of normative brain modeling. The main difference between brain age and normative modeling is that the latter uses the demographic variables to predict – a priori-defined – brain features. The degree to which our results can be extended to normative brain charts is discussed in detail in the revised version of the manuscript.

6) Discuss/clarify the effect size and the possible implications.

This issue is discussed in response to Reviewer #2, comment #3. All the main tests are well-powered to detect small effect sizes and use variables with high reliability. Reliability for longitudinal brain age delta is, however, unknown (and it has to be lower than cross-sectional brain age delta). Even assuming mediocre reliabilities, the tests would have enough power to detect small effect sizes – set aside that most research assumes a moderate-to-high relationship between cross-sectional brain age δ and brain aging. We now discuss the effect sizes and possible power issues in detail.

Reviewer #1 (Recommendations for the authors):Beyond what was summarized in the Public Review, here I note some errors in figures and text so that the authors can fix those, and suggestions for edits that should help with clarity.As already mentioned, the purpose of this study is to be applauded as mis- and over-interpretation of cross-sectional data findings as generalizing to true longitudinal change within individuals is a major problem in our field. The increasingly popular use of BrainAge as a metric has only increased this problem. The authors attempt to tests these assumptions was much needed.The format of the submission made the manuscript very difficult to parse (and I work immediately in this exact field). Primarily driving this difficult to understand the details of the paper are (1) the extreme brevity of the background/introduction and (2) the unfortunate reverse order of the manuscript sections -- it is always difficult when the results come before the methods or description of variables, but in this particular case, it was extremely disadvantageous. If this is the required journal format, then I guess not much van be done, but it is a hindrance to understanding the study. I found I could not read the Results section on its own and had to flip through to simultaneously read the Methods and Supplements. This was exhausting. Some readers may simply give up. One suggestion (highly recommended) is to utilize the Abstract to at least introduce some specifics about the study, like the variables to be used (morphological structural features instead of brain integrity or brain decline – no one can tell what the study will use as written). Similarly I did not know what polygenic scores were based on or what genetic amalgamation they represented until very far into the paper. Also portions of the intro were not clear on the first read, and only made sense after I finished the whole paper and supplemental info. Some word changes may help?

Thank you for the comment. We apologize that the manuscript was difficult to parse. The format of the journal specifies an Introduction-Results-Discussion-Methods format. However, the revised version of the manuscript includes your suggestions in the abstract and introduction sections as an attempt to increase readability. The abstract now defines the variables used (“brain structure features”), identifies the test and training datasets, and defines birth weight and polygenic scores of brain age as two congenital factors thought to reflect constant, lifelong influences emerging in early life. The introduction section also introduces specific references to the method and variables used to ease understanding of the results. See below for the main changes in both sections.

p.5 (Abstract): “Here, we explicitly tested this assumption in two independent large test datasets (UK Biobank [main] and Lifebrain [replication]; longitudinal observations ≈ 2,750 and 4,200)”

p.5 (Abstract): “Brain age models were estimated in two different training datasets (n ≈ 38,000 and 1,800 individuals [replication]) based on brain structural features.”

p.5 (Abstract): “Rather, brain age in adulthood was associated with the congenital factors of birth weight and polygenic scores of brain age, assumed to reflect a constant, lifelong influence on brain structure from early life.”

p.6 (Introduction): “Alternatively, individual deviations from the expected brain age could capture constant interindividual differences in brain structure that remain stable throughout the lifespan, reflecting early genetic and environmental influences (Deary, 2012; Elliott et al., 2019; Walhovd et al., 2016).”

p.6 (Introduction): “Here we tested whether brain age – derived from structural T1-weighted (T1w) morphological features – is related to accelerated brain aging, early-life factors, or a combination of both.”

p.7 (Introduction): “In addition, we also assessed brain change with a composite score of structural brain change as obtained using principal component analysis and change in the different raw structural brain features. These analyses were performed in two independent cohorts, both divided into a cross-sectional model generation (training) and a longitudinal, hypothesis testing (test) dataset.”

p.7 (Introduction): “If cross-sectional variations in brain age reflect differences in brain structure established early in life, one should observe a relationship between brain age and influences associated with stable, lifelong effects on brain structure. Here, we selected two congenital traits: self-reported birth weight and polygenic scores for brain age (PGS-BA), for which lifelong effects on age-related phenotypes have been shown (Walhovd et al., 2020, 2012) (Figure 1b).”

p.7 (Introduction): “Birth weight reflects normal variation in body (and brain) size as well as prenatal conditions, whereas PGS-BA quantifies genetic liability of having a higher brain age.”

Discussion: I thought that calling birth weight and genetics "early-life influences" throughout the paper a bit of an overstatement. Because this calls to mind many environmental things one experiences as a developing child/adolescent and was misleading (esp since things are not defined until very late in the sections, like Method at the end). A more accurate thing to call this would be something that made it clear that these variables represented "at birth indices". I do think this is a VERY important distinction and keeps the authors from leading the readers into a concept that is vague and perhaps not what is expected from that terminology.

Thank you for the comment. We agree it is important to disambiguate both terms and that “early-life influences” may be vague. We have replaced it with “congenital factors/indices'' as they refer to traits present from birth. However, both indices represent a proof-of-concept that interindividual variations in cross-sectional brain age delta reflect lifelong influences rooted in the distant past more than presently ongoing events, and thus to some extent can be generalizable to other non-studied factors that may exert a stable influence on age-related brain phenotypes rather than affect the slope of decline. The revised version disambiguates both concepts and discusses to which point the relationship between cross-sectional brain age and the congenital factors can be extrapolated to other congenital and early-life variables.

In p. 5 (Abstract): “Rather, brain age in adulthood was associated with the congenital factors of birth weight and polygenic scores of brain age, assumed to reflect a constant, lifelong influence on brain structure from early life.”

In p. 7 (Introduction): “If cross-sectional variations in brain age reflect differences in brain structure established early in life, one should observe a relationship between brain age and influences associated with stable, lifelong effects on brain structure. Here, we selected two congenital factors: self-reported birth weight and polygenic scores for brain age (PGS-BA), for which lifelong effects on age-related phenotypes have been shown (Walhovd et al., 2020, 2012) (Figure 1b).”

In p. 12 (Results): “Brain age delta is associated with congenital factors on brain structure”

In p. 15 (Discussion): “Rather, brain age seems to reflect early-life influences on brain structure, and only to a very modest degree reflects actual rate of brain change in middle and old adulthood. A lack of relationship between brain age and rate of brain aging can potentially be explained - although not investigated in the present study - by the effect of circumscribed events such as isolated insults or detrimental lifestyles that occurred in the past resulting in higher, but not accelerating, brain age. Yet, variations in brain age can equally reflect congenital and early-life differences and show lifelong stability.”

In p. 19 (Discussion): “Finally, many genetic and environmental factors relate to lifelong stable differences in brain age beyond birth weight and PGS-BA. However, both variables are congenital and show stable associations through the lifespan (Raznahan et al., 2012; Walhovd et al., 2020, 2016) without strong evidence that they relate to brain change after adolescence. Thus, birth weight and PGS-BA are paradigmatic for showing how interindividual differences in brain age emerge early in life. The present study does not provide a systematic understanding of these influences, but presents a framework for interpreting the impact such measures may exert on age-related phenotypes.”

Some study limitations are always warranted and a thoughtful inclusion. There really aren't any provided here to speak of. As a small example, I wondered what the impact is of very different sample sizes in different tests, but then comparisons across those analyses being made (as in the 770 for the birth weight, vs the 38k and 1372 for the other tests).

We now include a “Limitations” section. In this section, we discuss the following issues: (1) whether the analyses are well-powered; (2) the extent to which our results can be generalized to other normative modeling approaches and developmental samples (see Reviewer #2, comments #2, #4 for a wider discussion), and (3) whether the relationship between brain age and congenital factors PGS-BA and birth weight can be extended to other congenital and early life factors.

In pp. 17-9 (Discussion): “We used large training datasets to estimate the brain age models and the PGS scores leading to robust PGS-BA and brain age estimates. Self-reported birth weight (Nilsen et al., 2017) and cross-sectional brain age (Franke and Gaser, 2012) are highly reliable measures; thus, our analyses are well powered to detect small effects (Zuo et al., 2019). The reliability of brain age delta_long_ is, however, unknown. Strictly speaking, brain age delta is a prediction error from a model that maximizes the prediction of age in cross-sectional data and thus partially also reflects noise. Given that delta_long_ is estimated as the difference between two delta_cross_ estimates, it will hence have higher noise than the cross-sectional estimates reducing the power in identifying potential associations between longitudinal and cross-sectional delta; note also the relatively short interscan interval in UK Biobank (≈2y). However, our sample size (n > 1,200) ensures that the tests performed in this study are well-powered to detect small effects, even if delta_long_ has mediocre reliability (Zuo et al., 2019). Further, replication of our null results in the Lifebrain sample with more observations and longer follow-up times reduces the likelihood of noise as the main factor behind the lack of relationship. Furthermore, previous studies have found that changes in brain age are partly heritable (Brouwer et al., 2021) and relate to for instance cardiometabolic risk factors (Beck et al., 2021), suggesting that it captures biologically relevant signals (i.e. has predictive validity), although with substantially different origins from cross-sectional brain age. Although the reliability of delta_long_ needs to be formally tested, the null relationship between delta_cross_ and delta_long_ does not seem to be a result of a low-powered test.

We speculate that our results partially generalize to other normative and residual-based modeling approaches as well as to developmental samples. There is considerable evidence in the literature that birth weight and genetic risk for neurodegenerative conditions affects brain structure from early life (Raznahan et al., 2012; Walhovd et al., 2020, 2016, 2012b). Brain age models are related to other models such as normative brain charts (Bethlehem et al., 2021; Dong et al., 2020) - akin to normative anthropometric charts - the main difference being that brain age models predict, rather than control for, age (Marquand et al., 2019). Both types of models produce normative brain scores, which are uncorrelated with age (Butler et al., 2021). Thus, caution is required when interpreting these scores as indices of brain aging without availability of longitudinal data. Developmental samples may, however, reflect slightly stronger relationships between cross-sectional brain age delta and ongoing brain change as brain changes during early-life development typically occur at a faster pace than in middle or later life. Similarly, for specific disease groups such as Alzheimer’s disease patients (Franke and Gaser, 2012), interindividual brain variation in brain age might reflect to a greater extent prevailing loss of brain structure. Moreover, the variance associated with factors other than ongoing development/aging might be more limited in early than later age, since influences leading to interindividual variations in brain structure have a shorter span to accumulate. That is, as time from birth increases, chronological age as a marker of individual development is reduced.

Finally, many genetic and environmental factors relate to lifelong stable differences in brain age beyond birth weight and PGS-BA. However, both variables are congenital and show stable associations through the lifespan (Raznahan et al., 2012; Walhovd et al., 2020, 2016) without strong evidence that they relate to brain change after adolescence. Thus, birth weight and PGS-BA are paradigmatic for showing how interindividual differences in brain age emerge early in life. The present study does not provide a systematic understanding of these influences, but presents a framework for interpreting the impact such measures may exert on age-related phenotypes.”

Reviewer #2 (Recommendations for the authors):1. Measurement theory of individual differences should be included somewhere to clarify different focuses in terms of its two components: inter-individual (between-subject) variability and intra-individual (within-subject) variability. In theory, cross-sectional data is not enough to separate the two components and thus not good enough for any studies of individual differences, which call longitudinal data in ideal experimental design, to ensure both reliability and validity. Please see some comments in a recent publication (https://www.nature.com/articles/s41562-019-0655-x).

Thanks for the comment. Indeed, the relationship between cross-sectional and longitudinal brain age can be understood in the context of the measurement theory of individual differences (Brandmaier et al., 2018; Zuo et al., 2019). Previous work has shown that cross-sectional brain age is highly reliable (Franke and Gaser, 2012) but its validity – i.e. the proportion of the total variance attributed to the trait of interest alone, that is brain aging – has only been indirectly assessed. In this account, longitudinal brain change can be considered a “gold standard” criterion. We now discuss the findings under the framework of the measurement theory of individual differences.

p. 16 (discussion): “From a measurement theory perspective, our results suggest that cross-sectional brain age has low validity as an index of brain aging – despite having high reliability (Franke and Gaser, 2012) – as only a small portion of variance is associated with the trait of interest alone (Zuo et al., 2019). Most variance is rather associated with other factors that vary systematically across individuals, some of which are already present at birth.”

2. BrainAge is a recently developed method for normative aging research. What is the relationship between BrainAge and those metrics derived from the growth charts (e.g., height and weight).

Normative brain aging charts are analogous to normative anthropometric growth charts used in pediatrics. Brain age models can be considered a special case of normative brain modeling. The main difference between brain age and normative modeling is that the latter uses the demographic variables to predict a priori-defined brain features. Instead, Brain age models invert the approach, predicting age from several brain measures. Traditional normative charts are more easily interpretable than brain age whereas brain age models lead to simple outputs as they condense brain data into a single score (Marquand et al., 2019).

Both methods lead to scores that characterize the brain features of a given participant with respect to his/her peers (i.e. normative scores). Because age is invariably used in any normative model, both lead to brain measures that are uncorrelated with age (Butler et al., 2021). Researchers often interpret these measures as markers of ongoing brain change. We focused on brain age models as the interpretation of norm-deviation variations as accelerated/delayed aging is more pervasive, possibly due to:

a) Semantics, and that

b) The biological variables are selected and weighted based on their association with age. Thus, it is important to determine to which degree all these different metrics quantify advanced or delayed aging. Moreover, our call for caution interpreting these measures is generalizable. We now acknowledge the similarity between brain age models and normative brain models and discuss to what extent our word of caution can be extrapolated across models.

p. 18 (discussion): “We speculate that our results partially generalize to other normative and residual-based modeling approaches as well as to developmental samples. There is considerable evidence in the literature that birth weight and genetic risk for neurodegenerative conditions affect brain structure from early life (Raznahan et al., 2012; Walhovd et al., 2020, 2016, 2012b). Brain age models are related to other models such as normative brain charts (Bethlehem et al., 2021; Dong et al., 2020) – akin to normative anthropometric charts – the main difference being that brain age models predict, rather than control for, age (Marquand et al., 2019). Both types of models produce normative brain scores, which are uncorrelated with age (Butler et al., 2021). Thus, caution is required when interpreting these scores as indices of brain aging without proper assessment of longitudinal data.”

3. I noticed that the effect sizes detected are all small (e.g., the R2 < 0.01). In such large sample, how these weak associations are interpreted in terms of the measurement reliability and validity? What is the potential factors with impacts on the ability of detecting such small effect size?

Thank you for the question. Your assessment is correct. The significant relationships of birth weight and polygenic risk scores on brain age delta had an effect size of r2 ≈.009 and.02. The effect size of the non-significant relationship between cross-sectional and longitudinal brain age delta was r2 ≤ 0.001. We have little doubt these values stem from true positive and negative tests for the following reasons:

1) We used cross-sectional brain age delta, polygenic scores (PGS) of brain age, and self-reported birth weight as independent variables. Brain age δ and self-reported birth weight are highly reliable (Franke and Gaser, 2012; Nilsen et al., 2017); thus, not risking regression dilution effects.

2) We used cross-sectional and longitudinal brain age delta as dependent variables. The cross-sectional brain age delta has very high reliability. To our knowledge, there are no reports of the reliability of the longitudinal brain age delta. However, even if longitudinal brain age delta has a mediocre reliability for (≈ 0.4) – reliability for delta change certainly will be lower than for cross-sectional data – our test would still have enough power to detect small effects due to a relatively large sample size in both the main and the analyses (n > 1.200) (Zuo et al., 2019). The remaining tests use relatively large samples and involve measures with high reliability and thus are well-powered to detect very small effects.

Regarding the validity of the variables, longitudinal brain age delta has high validity as a measure of brain change while the validity of cross-sectional brain age delta as an index of brain aging is precisely a research question of the present study as discussed in Reviewer #1, comment #1. Likewise, the validity of PGS of brain age and self-reported birth weight as congenital factors is also high. Of course, both indices only capture a tiny fraction of the genetic and environmental influences that lead to interindividual differences in brain structure already in early life. Thus, the PGS and birth weight tests offer proof-of-concept that interindividual variance in brain age delta is more influenced by early life factors than by ongoing processes, rather than quantifying or approximating the total amount of variance in brain age delta that is explained by congenital factors. We now include a succinct summary of this explanation in the limitations section.

In pp 17-8 (discussion): “We used large training datasets to estimate the brain age models and the PGS scores leading to robust PGS-BA and brain age estimates. Self-reported birth weight (Nilsen et al., 2017) and cross-sectional brain age (Franke and Gaser, 2012) are highly reliable measures; thus, our analyses are well-powered to detect small effects (Zuo et al., 2019). The reliability of brain age delta_long_ is, however, unknown. Strictly speaking, brain age delta is a prediction error from a model that maximizes the prediction of age in cross-sectional data and thus partially also reflects noise. Given that delta_long_ is estimated as the difference between two delta_cross_ estimates, it will hence have higher noise than the cross-sectional estimates reducing the power in identifying potential associations between longitudinal and cross-sectional δ; note also the relatively short interscan interval in UK Biobank (≈2y). However, our sample size (n > 1,200) ensures that the tests performed in this study are well-powered to detect small effects, even if delta_long_ has mediocre reliability (Zuo et al., 2019). Further, replication of our null results in the Lifebrain sample with more observations and longer follow-up times reduces the likelihood of noise as the main factor behind the lack of relationship. Furthermore, previous studies have found that changes in brain age are partly heritable (Brouwer et al., 2021) and relate to for instance cardiometabolic risk factors (Beck et al., 2021), suggesting that it captures biologically relevant signals (i.e. has predictive validity), although with substantially different origins from cross-sectional brain age. Although the reliability of delta_long_ needs to be formally tested, the null relationship between delta_cross_ and delta_long_ does not seem to be a result of a low-powered test.”

In p. 19 (discussion): “Finally, many genetic and environmental factors relate to lifelong stable differences in brain age beyond birth weight and PGS-BA. However, both variables are congenital and show stable associations through the lifespan (Raznahan et al., 2012; Walhovd et al., 2020, 2016) without strong evidence that they relate to brain change after adolescence. Thus, birth weight and PGS-BA are paradigmatic for showing how interindividual differences in brain age emerge early in life. The present study does not provide a systematic understanding of these influences, but presents a framework for interpreting the impact such measures may exert on age-related phenotypes.”

4. This work is demonstrated for aging samples, but should be generalizable for developing samples. Please discuss about this point more comprehensively. Some recent work (e.g., https://www.sciencedirect.com/science/article/pii/S2095927320304965) may be of values.

Indeed. Both congenital factors (PGS of brain age δ and birth weight) will be associated with cross-sectional brain age delta in developmental samples. Previous research has found that other congenital indices such as preterm birth relate to brain age during childhood and adolescence. Other studies have found stable, lifelong effects of birth weight on brain structural features (Raznahan et al., 2012; Walhovd et al., 2012; Wheater et al., 2021), and stable effects of genetic factors of Alzheimer’s Disease on hippocampus volume during the entire lifespan (Walhovd et al., 2020).

Likewise, we also believe our caution when interpreting cross-sectional residual-based indices as accelerated/delayed maturation is extendable to developmental research. Yet, it is also likely that in development samples, brain age reflects ongoing changes to a greater degree than in aging for the following reasons: (1) During developmental interindividual differences in brain change are higher than in middle-age or aging; thus, changes in brain structure are generally steeper in development than in aging. This feature might also apply to specific disease groups in aging such as in Alzheimer’s disease patients. (2) Variance associated with variables other than ongoing brain change that systematically vary across individuals, should be lower than in older adults where a longer lifespan should lead to a wider and higher accumulation of effects on the individuals’ brain structure. Thus, as time from birth increases, chronological age as a marker of individual development is reduced. Whereas present knowledge allows us to extend our call for caution to a developmental context, the degree to which brain age delta reflects ongoing development in younger samples, needs to be formally tested. The revised version of the manuscript includes a brief version of the present argumentation.

In p. 18 (discussion): “We speculate that our results partially generalize to other normative and residual-based modeling approaches as well as to developmental samples. There is considerable evidence in the literature that birth weight and genetic risk for neurodegenerative conditions affects brain structure from early life (Raznahan et al., 2012; Walhovd et al., 2020, 2016, 2012b). Brain age models are related to other models such as normative brain charts (Bethlehem et al., 2021; Dong et al., 2020) – akin to normative anthropometric charts – the main difference being that brain age models predict, rather than control for, age (Marquand et al., 2019). Both types of models produce normative brain scores, which are uncorrelated with age (Butler et al., 2021). Thus, caution is required when interpreting these scores as indices of brain aging without availability of longitudinal data. Developmental samples may, however, reflect slightly stronger relationships between cross-sectional brain age delta and ongoing brain change as brain changes during early-life development typically occur at a faster pace than in middle or later life. Similarly, for specific disease groups such as Alzheimer’s disease patients (Franke and Gaser, 2012), interindividual brain variation in brain age might reflect to a greater extent prevailing loss of brain structure. Moreover, the variance associated with other factors than ongoing aging/development might be more limited in early than later age as factors leading to interindividual variations in brain structure have a shorter span to accumulate. That is, as time from birth increases, chronological age as a marker of individual development is reduced.”

References

Beck D, Lange A-MG de, Pedersen ML, Alnæs D, Maximov II, Voldsbekk I, Richard G, Sanders A-M, Ulrichsen KM, Dørum ES, Kolskår KK, Høgestøl EA, Steen NE, Djurovic S, Andreassen OA, Nordvik JE, Kaufmann T, Westlye LT. 2021. Cardiometabolic risk factors associated with brain age and accelerate brain ageing. *medRxiv* 2021.02.25.21252272. doi:10.1101/2021.02.25.21252272

Bethlehem R a. I, Seidlitz J, White SR, Vogel JW, Anderson KM, Adamson C, Adler S, Alexopoulos GS, Anagnostou E, Areces-Gonzalez A, Astle DE, Auyeung B, Ayub M, Ball G, Baron-Cohen S, Beare R, Bedford SA, Benegal V, Beyer F, Bae JB, Blangero J, Cábez MB, Boardman JP, Borzage M, Bosch-Bayard JF, Bourke N, Calhoun VD, Chakravarty MM, Chen C, Chertavian C, Chetelat G, Chong YS, Cole JH, Corvin A, Courchesne E, Crivello F, Cropley VL, Crosbie J, Crossley N, Delarue M, Desrivieres S, Devenyi G, Biase MAD, Dolan R, Donald KA, Donohoe G, Dunlop K, Edwards AD, Elison JT, Ellis CT, Elman JA, Eyler L, Fair DA, Fletcher PC, Fonagy P, Franz CE, Galan-Garcia L, Gholipour A, Giedd J, Gilmore JH, Glahn DC, Goodyer I, Grant PE, Groenewold NA, Gunning FM, Gur RE, Gur RC, Hammill CF, Hansson O, Hedden T, Heinz A, Henson R, Heuer K, Hoare J, Holla B, Holmes AJ, Holt R, Huang H, Im K, Ipser J, Jack CR, Jackowski AP, Jia T, Johnson KA, Jones PB, Jones DT, Kahn R, Karlsson H, Karlsson L, Kawashima R, Kelley EA, Kern S, Kim K, Kitzbichler MG, Kremen WS, Lalonde F, Landeau B, Lee S, Lerch J, Lewis JD, Li J, Liao W, Linares DP, Liston C, Lombardo MV, Lv J, Lynch C, Mallard TT, Marcelis M, Markello RD, Mazoyer B, McGuire P, Meaney MJ, Mechelli A, Medic N, Misic B, Morgan SE, Mothersill D, Nigg J, Ong MQW, Ortinau C, Ossenkoppele R, Ouyang M, Palaniyappan L, Paly L, Pan PM, Pantelis C, Park MM, Paus T, Pausova Z, Binette AP, Pierce K, Qian X, Qiu J, Qiu A, Raznahan A, Rittman T, Rollins CK, Romero-Garcia R, Ronan L, Rosenberg MD, Rowitch DH, Salum GA, Satterthwaite TD, Schaare HL, Schachar RJ, Schultz AP, Schumann G, Schöll M, Sharp D, Shinohara RT, Skoog I, Smyser CD, Sperling RA, Stein DJ, Stolicyn A, Suckling J, Sullivan G, Taki Y, Thyreau B, Toro R, Tsvetanov KA, Turk-Browne NB, Tuulari JJ, Tzourio C, Vachon-Presseau É, Valdes-Sosa MJ, Valdes-Sosa PA, Valk SL, Amelsvoort T van, Vandekar SN, Vasung L, Victoria LW, Villeneuve S, Villringer A, Vértes PE, Wagstyl K, Wang YS, Warfield SK, Warrier V, Westman E, Westwater ML, Whalley HC, Witte AV, Yang N, Yeo BTT, Yun HJ, Zalesky A, Zar HJ, Zettergren A, Zhou JH, Ziauddeen H, Zugman A, Zuo XN, AIBL, Initiative ADN, Investigators ADRWB, ASRB, Team C, Cam-CAN, Ccnp 3r-Brain, COBRE, Group EDBA working, FinnBrain, Study HAB, Imagen K, NSPN, OASIS-3, Project O, POND, The PREVENT-AD Research Group V, Alexander-Bloch AF. 2021. Brain charts for the human lifespan. *bioRxiv* 2021.06.08.447489. doi:10.1101/2021.06.08.447489

Brandmaier AM, Wenger E, Bodammer NC, Kühn S, Raz N, Lindenberger U. 2018. Assessing reliability in neuroimaging research through intra-class effect decomposition (ICED). *eLife* 7:e35718. doi:10.7554/*eLife*.35718

Brouwer RM, Schutte J, Janssen R, Boomsma DI, Hulshoff Pol HE, Schnack HG. n.d. The Speed of Development of Adolescent Brain Age Depends on Sex and Is Genetically Determined. *Cereb Cortex*. doi:10.1093/cercor/bhaa296

Butler ER, Chen A, Ramadan R, Le TT, Ruparel K, Moore TM, Satterthwaite TD, Zhang F, Shou H, Gur RC, Nichols TE, Shinohara RT. 2021. Pitfalls in brain age analyses. *Human Brain Mapping* 42:4092–4101. doi:10.1002/hbm.25533

Deary IJ. 2012. Looking for “system integrity” in cognitive epidemiology. *Gerontology* 58:545–553. doi:10.1159/000341157

Dong H-M, Castellanos FX, Yang N, Zhang Z, Zhou Q, He Y, Zhang L, Xu T, Holmes AJ, Thomas Yeo BT, Chen F, Wang B, Beckmann C, White T, Sporns O, Qiu J, Feng T, Chen A, Liu X, Chen X, Weng X, Milham MP, Zuo X-N. 2020. Charting brain growth in tandem with brain templates at school age. *Science Bulletin* 65:1924–1934. doi:10.1016/j.scib.2020.07.027

Elliott ML, Belsky DW, Knodt AR, Ireland D, Melzer TR, Poulton R, Ramrakha S, Caspi A, Moffitt TE,

Hariri AR. 2019. Brain-age in midlife is associated with accelerated biological aging and cognitive decline in a longitudinal birth cohort. *Mol Psychiatry*. doi:10.1038/s41380-019-0626-7

Franke K, Gaser C. 2012. Longitudinal changes in individual BrainAGE in healthy aging, mild cognitive impairment, and Alzheimer’s disease. GeroPsych: The Journal of Gerontopsychology and Geriatric Psychiatry 25:235–245. doi:10.1024/1662-9647/a000074

Marquand AF, Kia SM, Zabihi M, Wolfers T, Buitelaar JK, Beckmann CF. 2019. Conceptualizing mental disorders as deviations from normative functioning. *Mol Psychiatry* 24:1415–1424. doi:10.1038/s41380-019-0441-1

Nilsen TS, Kutschke J, Brandt I, Harris JR. 2017. Validity of Self-Reported Birth Weight: Results from a Norwegian Twin Sample. *Twin Res Hum Genet* 20:406–413. doi:10.1017/thg.2017.44

Raznahan A, Greenstein D, Lee NR, Clasen LS, Giedd JN. 2012. Prenatal growth in humans and postnatal brain maturation into late adolescence. *PNAS* 109:11366–11371. doi:10.1073/pnas.1203350109

Walhovd KB, Fjell AM, Brown TT, Kuperman JM, Chung Y, Hagler DJ, Roddey JC, Erhart M, McCabe C, Akshoomoff N, Amaral DG, Bloss CS, Libiger O, Schork NJ, Darst BF, Casey BJ, Chang L, Ernst TM, Frazier J, Gruen JR, Kaufmann WE, Murray SS, van Zijl P, Mostofsky S, Dale AM, Pediatric Imaging, Neurocognition, and Genetics Study. 2012. Long-term influence of normal variation in neonatal characteristics on human brain development. *Proc Natl Acad Sci USA* 109:20089–20094. doi:10.1073/pnas.1208180109

Walhovd KB, Fjell AM, Sørensen Ø, Mowinckel AM, Reinbold CS, Idland A-V, Watne LO, Franke A, Dobricic V, Kilpert F, Bertram L, Wang Y. 2020. Genetic risk for Alzheimer disease predicts hippocampal volume through the human lifespan. *Neurology Genetics* 6. doi:10.1212/NXG.0000000000000506

Walhovd KB, Krogsrud SK, Amlien IK, Bartsch H, Bjørnerud A, Due-Tønnessen P, Grydeland H, Hagler DJ, Håberg AK, Kremen WS, Ferschmann L, Nyberg L, Panizzon MS, Rohani DA, Skranes J, Storsve AB, Sølsnes AE, Tamnes CK, Thompson WK, Reuter C, Dale AM, Fjell AM. 2016. Neurodevelopmental origins of lifespan changes in brain and cognition. *Proc Natl Acad Sci USA* 113:9357–9362. doi:10.1073/pnas.1524259113

Wheater E, Shenkin SD, Muñoz Maniega S, Valdés Hernández M, Wardlaw JM, Deary IJ, Bastin ME, Boardman JP, Cox SR. 2021. Birth weight is associated with brain tissue volumes seven decades later but not with MRI markers of brain ageing. *NeuroImage: Clinical* 31:102776. doi:10.1016/j.nicl.2021.102776

Zuo X-N, Xu T, Milham MP. 2019. Harnessing reliability for neuroscience research. *Nat Hum Behav* 3:768–771. doi:10.1038/s41562-019-0655-x